# Detrimental Roles of Hypoxia-Inducible Factor-1α in Severe Hypoxic Brain Diseases

**DOI:** 10.3390/ijms25084465

**Published:** 2024-04-18

**Authors:** Yoon Kyung Choi

**Affiliations:** Department of Integrative Bioscience and Biotechnology, Konkuk University, Seoul 05029, Republic of Korea; ykchoi@konkuk.ac.kr; Tel.: +82-2-450-0558

**Keywords:** cell death, central nervous system, hypoxia-inducible factor, inflammasome, mitochondria, severe hypoxia

## Abstract

Hypoxia stabilizes hypoxia-inducible factors (HIFs), facilitating adaptation to hypoxic conditions. Appropriate hypoxia is pivotal for neurovascular regeneration and immune cell mobilization. However, in central nervous system (CNS) injury, prolonged and severe hypoxia harms the brain by triggering neurovascular inflammation, oxidative stress, glial activation, vascular damage, mitochondrial dysfunction, and cell death. Diminished hypoxia in the brain improves cognitive function in individuals with CNS injuries. This review discusses the current evidence regarding the contribution of severe hypoxia to CNS injuries, with an emphasis on HIF-1α-mediated pathways. During severe hypoxia in the CNS, HIF-1α facilitates inflammasome formation, mitochondrial dysfunction, and cell death. This review presents the molecular mechanisms by which HIF-1α is involved in the pathogenesis of CNS injuries, such as stroke, traumatic brain injury, and Alzheimer’s disease. Deciphering the molecular mechanisms of HIF-1α will contribute to the development of therapeutic strategies for severe hypoxic brain diseases.

## 1. Introduction

Severe hypoxia affects the central nervous system (CNS) by triggering neurovascular inflammation, oxidative stress, glial activation, impaired mitochondrial function, and cell death [1]. High O_2_ therapy can elevate cerebral blood flow and improve cognitive behavioral performance by diminishing hypoxia involved in the pathogenesis of Alzheimer’s disease (AD) [2,3]. Hypoxia-inducible factors (HIF1–3) regulate transcriptional responses to reduce O_2_ availability [4]. HIFs are heterodimeric proteins that are composed of an O_2_-regulated HIF-α subunit and a constitutively expressed HIF-1β subunit. HIF-α subunits are subject to prolyl hydroxylation, which targets proteins for degradation under normoxic conditions [5,6]. Two HIF-α proteins, HIF-1α and HIF-2α, are stabilized under low O_2_ tension and dimerize with HIF-1β. Heterodimeric proteins bind to hypoxia-responsive elements in multiple target genes and regulate their transcription to facilitate adaptation to hypoxia [7].

HIF-1α accumulation has dual effects, including cell death and cell survival, in neurovascular diseases such as stroke, traumatic brain injury (TBI), and AD [8,9,10,11,12,13,14]. HIF-1α has complex effects in the brain [15,16,17], which largely depend on the severity of and time-point after hypoxic damage. HIF-1α stabilization during mild hypoxia may enhance cell regeneration (i.e., angiogenesis and neurogenesis), mitochondrial biogenesis, and cell survival in the brain through HIF-1α target genes [16,17]. In severe hypoxia, HIF-1α causes various gene expression changes and post-translational modifications related to cell damage, mitochondrial dysfunction, cellular lipid peroxidation, and inflammasome formation [18,19,20,21,22] (Figure 1). Here, this review focuses on the detrimental effects of HIF-1α on cell damage under severe hypoxic conditions.

Narrowing or blockage of arteries can induce ischemic stroke, leading to reactive oxygen species (ROS)-mediated death of neurons, endothelial cells, and glia including oligodendrocytes [23,24]. TBI is an acquired brain injury caused by a mechanical impact on the head [25]. Individuals with severe ischemia or trauma are more susceptible to the development of AD [26,27]. Some cases of dementia may arise from cerebral hypoperfusion after ischemic injury due to decreased beta-amyloid (Aβ) clearance or catabolism [26,27,28]. A significant increase in microglia-specific thromboxane A synthase 1 was observed in the human AD brain [29]. Notably, thromboxane A is a potent vasoconstrictor in the cerebral circulation and is also a target for the secondary prevention of stroke [30]. Therefore, hypoxia-related diseases may share similar pathological pathways.

This review discusses the current evidence regarding the contribution of severe hypoxia to CNS injuries, with an emphasis on HIF-1α-mediated pathways. Understanding the role of these pathways in severe hypoxic CNS injuries, such as ischemic stroke, TBI, and AD, will provide clues for therapeutic strategies. HIF-1α-mediated pathways include neurovascular inflammation, oxidative stress, glial activation, vascular damage, mitochondrial dysfunction, and cell death (Figure 1) [1,14,22,31]. This review describes the important actions of HIF-1α in severe hypoxic CNS injuries and their potential pathogenetic mechanisms.

## 2. Role of HIF in Cell Damage

Several cell death pathways are associated with severe hypoxia. This section discusses the roles of HIF-1α in types of cell death in the CNS. 

### 2.1. Apoptosis

Apoptosis refers to the process of programmed cell death, characterized by the orchestrated collapse of a cell, with membrane blebbing, cell shrinkage, chromatin condensation, and DNA fragmentation [32]. While many reports have demonstrated the protective effects of HIF-1α on metabolic adaptation during mild hypoxia, activation of the HIF-1α signaling pathway during severe hypoxia is associated with cell death [15]. Genetic neuronal HIF-1α and HIF-2α deficiencies triggered neuronal survival and sensorimotor function in an ischemic stroke model [8]. BCL-2/adenovirus E1B 19 kDa-interacting protein 3 (BNIP3), a downstream target gene of HIF-1α, is involved in apoptosis [15,33]. HIF-1 is linked to oxidative stress-induced Aβ accumulation and subsequent activation of the pro-death gene *BNIP3* in primary cortical neurons [34]. In TBI, HIF-1α mediates tumor necrosis factor (TNF)-related apoptosis, inducing ligand-induced neuronal apoptosis [35]. Pericyte cell death is related to HIF-1α-mediated caspase 3 activation during TBI [9].

Apoptosis is related to mitochondrial permeability. The release of mitochondrial DNA (mtDNA) during mitochondrial outer membrane permeabilization generally involves the pro-apoptotic pore-formation proteins BCL-2-associated X, apoptosis regulator (BAX), and BCL-2 antagonist/killer 1 (BAK1) [36]. The relative availability of BAX and BAK molecules tunes apoptotic pore growth to control mtDNA-mediated inflammation [37]. BAX–BAK1 pores in the outer mitochondrial membrane enable extrusion of the inner mitochondrial membrane into the cytosol, culminating in inner mitochondrial membrane breakdown and cytosolic release of mtDNA [38]. A positive relationship between HIF-1α and pro-apoptotic proteins, such as BAX, has been reported [18,39]. Overexpression of HIF-1α upregulates active monomers of BAX in hypoxic cardiomyocyte cells [18]. HIF-1α modulates apoptosis-inducing proteins (i.e., BAX, BNIP3), leading to mitochondrial dysfunction and consequent cell death under severe hypoxic conditions [40,41].

### 2.2. Ferroptosis

Ferroptosis is an iron-dependent form of cell death that results from increased ROS and lipid peroxidation in the plasma and mitochondrial membranes [42]. The Fenton reaction involves the production of ROS from the reaction between H_2_O_2_ and ferrous iron (Fe^2+^), which can trigger lipid peroxidation. HIF-1α plays a role in CNS ferroptosis. During hypoxia, HIF-1α upregulates heme oxygenase-1 (HO-1, encoded by *HMOX1*) gene expression [43,44]. HO-1 resides within the endoplasmic reticulum, and its sustained expression in glial fibrillary acidic protein (GFAP)-expressing astrocytes exacerbates AD development [44]. Transgenic mice exhibiting prolonged expression of HO-1 in astrocytes (GFAP.HMOX1 transgenic) acquire abnormal iron deposition in the mitochondria of astrocytes located in the striatum as well as neuronal deficiency and reduced cognitive ability [45,46,47,48]. When GFAP.HMOX1 transgenic mice-derived astrocytes and neurons are co-cultured, active caspase-3 (an apoptotic factor) is increased in transgenic-derived neurons compared with wild-type preparations [48]. These studies suggest that overexpression of HO-1 in astrocytes may give rise to neuronal dysfunction. 

Mitoferrin 2 (MFRN2) is an iron transporter found in the mitochondrial membrane. *MFRN2* gene silencing inhibits mitochondrial iron overload and stabilizes the mitochondrial membrane potential in TNF-α-treated endothelial cells [49]. Abnormal iron accumulation is associated with cognitive impairment in patients with AD [50]. Induction of ferroptosis by excessive HO-1 overexpression may be associated with the reduction in the free iron-binding ability of ferritin induced by pro-oxidant conditions (i.e., heme, Aβ, H_2_O_2_, dopamine, hyperoxia, ultraviolet light, heavy metals, prostaglandins, and nitric oxide) [51,52]. HO-1 accelerates erastin-induced ferroptotic death in fibrosarcoma cells [51]. Induction of ferroptosis by the HIF-1α/HO-1 pathway has been reported in various cells, including neuronal and vascular smooth muscle cells [19,20,21]. Ferroptosis aggravated diabetic nephropathy and damaged renal tubules via the HIF-1α/HO-1 pathway in a mouse model of diabetes [53]. Compared to sodium iodate (an ROS inducer) alone, sodium iodate with hypoxia-mediated HIF-1α stabilization markedly increases retinal pigment epithelial cell death via ferroptosis [54].

BNIP3 can act as the upstream regulator of the HIF-1α-mediated glycolytic program [55]. In melanoma cells, BNIP3 deficiency results in increased intracellular iron levels caused by heightened nuclear receptor coactivator 4 (NCOA4)-mediated autophagic degradation of ferritin (ferritinophagy), which facilitates HIF-1α degradation [55]. NCOA4 is a selective cargo receptor that mediates autophagic degradation of ferritin, a cytosolic iron-storage complex [56]. Under autophagy-disrupting conditions, NCOA4 may be unable to target ferritin for lysosomal degradation, resulting in the accumulation of free iron [57]. Both HIF-1α and HIF-2α upregulate NCOA4 expression in hepatic cells treated with the iron chelator deferoxamine [58]. Taken together, HIF-1α may upregulate iron production and mitochondrial accumulation via HO-1. Additionally, HIF-α may be involved in ferritinophagy through the NCOA4-mediated pathway. 

## 3. Role of HIF in Inflammasomes 

Pyroptosis refers to an inflammatory form of cell death involving caspase-1 activation and consequent cleavage of the pore-forming protein gasdermin D [59]. Inflammation can be regulated by the nucleotide-binding domain and leucine-rich-repeat-containing receptor (NLR) family, to form large multiprotein complexes called inflammasomes [60]. Inflammasomes include NLR family pyrin domain-containing 1 (NLRP1), NLRP3, NLRP6, NLRP7, NLRP12, caspase activation and recruitment domain (CARD)-domain containing 4 (NLRC4), and absent in melanoma 2 (AIM2) [61,62]. 

NLRP3 interacts with the adaptor molecule apoptosis-associated speck-like protein-containing CARD (ASC) via its pyrin domain (PYD). The CARD domain of ASC recruits the CARD domain of pro-caspase-1 to form the NLRP3–ASC–pro-caspase-1 complex, which is also known as the NLRP3 inflammasome [63,64,65]. Upon activation, NLRP1 directly interacts with pro-caspase-1, without the adaptor protein ASC [66]. However, ASC can also increase NLRP1-mediated caspase-1 activation [67]. Double-stranded DNA (dsDNA) from the nucleus and mitochondria induces AIM2 activation [68]. The AIM2 inflammasome increases the conversion of gasdermin D into gasdermin D-N fragments, which leads to pyroptosis [68]. The observation that AIM2 overexpression results in greater gasdermin D activity, and vice versa, in atherosclerotic plaques of apolipoprotein E^−/−^ (ApoE^−/−^) mice further supports the role of AIM2 in mediating pyroptotic cell death [69].

Caspases belong to the cysteine-dependent protease family and play key roles in various forms of cell damage [70]. Caspase-3, -6, -7, -8, and -9 are involved in apoptosis in mammals. Caspase-1, -4, -5, and -12 are associated with inflammation in humans [71]. CoCl_2_-induced HIF-1α induces apoptosis by upregulating caspase-3, -8, and -9 in human fibroblasts [72]. Retinal ischemia/reperfusion increases Toll-like receptor 4 (TLR4) expression, triggering caspase-8 signaling [73]. Caspase-8 promotes NLRP1 and NLRP3 inflammasome activation and interleukin (IL)-1β production in an acute glaucoma model [73]. Intravitreous injection of a caspase-8 inhibitor reduces ischemia/reperfusion-induced NLRP1 and NLRP3 inflammasomes and IL-1β production [73]. Hypoxic injury is associated with inflammasome formation in the brain. In the following sections, the relationship between HIF-1α and inflammasomes in neurovascular diseases, such as stroke, TBI, and AD, is discussed. 

### 3.1. Role of HIF in Stroke, with a Focus on Inflammasomes

Higher HIF-1α levels have been significantly correlated with the initial stroke scale score, indicating a worse outcome [13]. Necrosis is an accidental cell death that results in the uncontrolled release of inflammatory cellular contents [74]. Necroptosis mimics features of both apoptosis and necrosis. Necroptosis requires related proteins, such as receptor-interacting protein kinase-3 (RIPK3) and the effector mixed lineage kinase domain-like protein (MLKL) [75]. HIF-1α also regulates necroptosis-related proteins, such as RIPK3 and MLKL, in ischemic stroke [75]. Enhanced HIF-1α levels after ischemic stroke appear to be involved in RIPK3/MLKL activation, leading to activation of the NLRP3 inflammasome [75]. HIF-1α induces NLRP3 inflammasome-dependent pyroptotic and apoptotic cell death following ischemic stroke in adult rats [76]. Treatment with an HIF-1α inhibitor reduces macrophage and neutrophil infiltration in the ipsilateral brain [76]. 

The role of NLRP1 in acute ischemic brain injury has also been previously demonstrated [77]. Nuclear factor-κB (NF-κB) and mitogen-activated protein kinase (i.e., p38, JNK, and ERK) inhibitors attenuate the expression and activation of inflammasome-related proteins, such as NLRP1, NLRP3, ASC, IL-1β, and IL-18, in the brain after focal ischemic stroke [78]. In neuronal cells exposed to oxygen–glucose deprivation, increased levels of inflammasome-related proteins (i.e., NLRP1, ASC, caspase-1, IL-1β, and IL-18) were significantly reduced by microRNAs (i.e., miR-9a-5p) [79]. miR-9a-5p binds within the 3′-untranslated region of *NLRP1* [79]. Thus, the downregulation of miR-9a-5p enhances NLRP1 inflammasome-mediated ischemic injury by upregulating NLRP1 expression [79]. 

In a mouse model of cerebral ischemia, AIM2 and NLRC4 inflammasomes, along with ASC, contributed to the development of acute brain injury [61]. Chronic cerebral hypoperfusion activates and upregulates the AIM2 and NLRP3 inflammasomes [80]. The expression of NLRP3 and AIM2 is upregulated in glial cells in the brains of patients with cerebral infarction in the chronic phase, suggesting that chronic cerebral hypoperfusion induces inflammasomes [80]. 

### 3.2. Role of HIF in TBI, with a Focus on Inflammasomes

HIF-1α aggravates TBI via NLRP3-inflammasome-mediated pyroptosis and microglial activation by 3 days after TBI [22]. NLRP3 is mainly expressed in microglia and has also been detected in endothelial cells, astrocytes, and oligodendrocytes [81,82]. Administration of an HIF-1α inhibitor to TBI model mice reduces TBI-mediated NLRP3 protein levels and blood–brain barrier (BBB) breakdown, showing improvement of behavioral functions [22]. In the TBI brain, the levels of NLRP1, NLRP3, NLRC4, and AIM2 were found to be increased in microvascular endothelial cells [82]. Administration of a caspase-1 inhibitor after TBI decreases pyroptosis, as evidenced by decreased cleaved gasdermin D and IL-1β levels, and alleviates TBI-induced BBB leakage without affecting the expression of NLRP1, NLRP3, NLRC4, and AIM2 [82]. Compared to wild-type mice, cortical samples of Nlrp1^−^ and Asc^−/−^ mice that had been subjected to TBI showed reduced levels of proinflammatory cytokines, such as IL-1β and IL-6 [83]. However, motor deficits did not change in Nlrp1^−/−^ and Asc^−/−^ mice as compared to wild-type mice after TBI [83]. Further studies revealing the relationship between inflammasome blockade and the improvement in behavioral function after TBI are required. 

### 3.3. Role of HIF in AD, Focusing on Inflammasomes

Hypoxia facilitated plaque formation in an AD transgenic mouse model, leading to memory deficits [84]. Various pathogenic mechanisms of AD have been considered, including chronic hypoxia, amyloid precursor protein (APP) expression, Aβ aggregation, and hyperphosphorylated tau protein accumulation [1,2,85]. Chronic hypoxia-mediated HIF-1α may upregulate the activity of β-site APP-cleaving enzyme 1, facilitate the β-cleavage of APP, increase Aβ deposition, and potentiate memory deficits in APP23 transgenic AD mice [1,84,86]. In addition, HIF-1α binds to the γ-secretase subunit gene promoter and subsequently induces γ-secretase-mediated Aβ production during hypoxia [87]. NLRP1 and NLRP3 are more greatly activated in monocytes obtained from patients with AD than in those from healthy controls [88]. Mitochondrial ROS drive the assembly of the NLRP3 inflammasome inside microglia [89], consequently facilitating tau pathology [90]. The transfer of miR-146a-5p from microglial exosomes into intermittently hypoxic neurons reduces the mRNA levels of HIF-1α, NLRP3, IL-1β, and IL-18 [91]. These effects are reversed by overexpression of HIF-1α [91]. Chronic intermittent hypoxia induces the formation of the neuronal NLRP3 inflammasome, which is a key regulator of neuroinflammation during cognitive impairment [91]. HIF-1α and NLRP1 protein levels were markedly increased in the endothelial cells of the brain and the retinas of triple (PS1M146V, APPswe, and tauP301L) transgenic mouse models of AD aged 16 months [14]. In this AD mouse model, HIF-1α protein expression was observed in the cytoplasm of endothelial cells in the brain and retina [14]. Cytoplasmic accumulation of HIF-1α may have been related to apoptosis and necrosis in the cerebral cortexes of 24-month-old rats exposed to intermittent hypoxia [92]. During oxygen–glucose deprivation, endothelial cells show upregulated HIF-1α and NLRP1 protein levels, while downregulation of HIF-1α reduces NLRP1 expression, and vice versa [14]. Thus, HIF-1α can affect inflammasomes in sustained hypoxic injuries. 

## 4. Role of HIF-1α in Mitochondrial Functions

During hypoxia, ROS are generated due to mitochondrial depolarization especially through complex I and III, activation of xanthine oxidase, and NADPH oxidases at different oxygen levels in the brain [93]. Mitochondria consume O_2_ for ATP production. Hypoxia elicits mitochondrial ROS accumulation due to the insufficient number of electrons supplied by O_2_, causing imbalanced electron transfer in the electron transport chain [94]. Mitochondrial dysfunction is the main cause of energy failure in damaged tissues and is the basis for cell death. Activation of mitochondrial ATP-sensitive K^+^ channels promotes HIF-1α expression in ischemic injuries [95,96]. A positive feedback regulation between ROS production and the HIF pathway has also been reported [97,98]. ROS and reactive nitrogen species (RNS) contribute to oxidative stress production [99]. One RNS, peroxynitrite (ONOO^−^), can be formed by nitric oxide (NO) and O_2_^−^ [100]. Hypoxia-induced HIF-1α can upregulate inducible nitric oxide synthase (iNOS) expression [101], leading to NO production. Uncoupling of endothelial NOS produces O_2_^−^ during hypoxia [102]. 

The repair strategy for hypoxic neurovascular diseases involves artificial mitochondrial transfer/transplantation by transferring healthy mitochondria into damaged cells. Mitochondrial transplantation is an emerging therapeutic approach for the treatment of hypoxic neurovascular diseases [103,104,105,106]. Transfer of healthy mitochondria ameliorates cognitive deficits and neuronal damage and increases cell viability [103,104,105,106]. Mitochondrial transfer holds great potential for maintaining homeostasis during pathological processes.

### 4.1. Mitochondrial DNA

The released mtDNA, cytosolic double-stranded DNA, can act as a ligand for various detrimental signal sensors, activating an innate immune response in a caspase-independent manner [107]. These signal sensors include the NLRP3 inflammasome, AIM2 inflammasome, and the cytosolic cyclic GMP–AMP synthase (cGAS)-stimulator of interferon genes (STING) pathway [68,70,107]. cGAS-STING signaling cascades facilitate the mtDNA-mediated secretion of inflammatory cytokines and activate immune cells [70,107] (Figure 2). mtDNA can be oxidized by ROS to generate fragments [107,108]. The binding of cytosolic oxidized mtDNA to the NF-κB-mediated NLRP3 inflammasome suggests a link between apoptosis and the inflammasome [109]. Autophagy proteins (i.e., microtubule-associated protein 1 light chain 3B [LC3B] and beclin1) regulate the innate immune response by inhibiting NLRP3-inflammasome-mediated mtDNA release, leading to the preservation of mitochondrial integrity [110]. Autophagy is an evolutionarily conserved, lysosome-dependent mechanism through which eukaryotic cells eliminate potentially cytotoxic or superfluous materials from the cytoplasm, thereby maintaining homeostasis [70]. Depletion of autophagic proteins promotes the accumulation of dysfunctional mtDNA in the cytosol in response to lipopolysaccharides and ATP in macrophages [110]. CoCl_2_-induced hypoxia upregulates the expression of autophagy-related genes (ATGs), such as *BNIP3*, *BECN1*, *LC3*, *ATG5*, and *ATG7* [111]. HIF-1α inhibitors reduce hypoxic preconditioning-mediated enhancement of BNIP3 and beclin1 protein levels [112].

### 4.2. HIF-1α–BNIP3 Axis in Mitochondrial Functions

HIF-1α regulates BNIP3 in various cells during hypoxia; however, the HIF-1α–BNIP3 axis can be beneficial or detrimental in a cell-type-dependent manner. Recent reports have shown that HIF-1α and BNIP3 can be translocated to the mitochondria [113,114]. Mitochondrial HIF-1α may play protective roles by reducing ROS generation in response to hypoxia [115]. Under hypoxia/reoxygenation (H/R) conditions, HIF-1α–BNIP3-mediated mitophagy protects tubular cells from renal injury [116]. *HIF1A* knockout attenuates H/R-induced mitophagy and aggravates H/R-induced apoptosis; these effects are reversed by BNIP3 overexpression in acute kidney damage [116]. Mitophagy is an autophagic response that preferentially degrades permeabilized or otherwise dysfunctional mitochondria [70]. In various hypoxic cells, the signal transducer and activator of transcription 3 (STAT3) and HIF-1α cooperate in the nucleus to transcribe HIF-1α target genes, such as vascular endothelial growth factor (*VEGF*) and hexokinase1 (*HK1*) [117,118]. STAT3 transcriptionally activates target genes, including *HIF1A*, possibly increasing BNIP3 expression [119]. Mitochondrial translocation of STAT3 suppresses autophagy induced by oxidative stress [113]. STAT3 may protect the mitochondria from degradation by mitophagy [113]. 

Mitochondrial damage may enhance the HIF-1α–BNIP3 axis, thereby promoting mitophagy during hypoxia [120]. Hippocampal neurons show hypoxia- and aging-associated disruption of mitochondrial cristae (reviewed in [121]), possibly resulting in mitophagy. Mitophagy can be evaluated by measuring the mtDNA copy number and changes in mitophagy-related proteins, including translocase of the outer mitochondrial membrane complex subunit 20, cytochrome c oxidase IV, LC3B, and the mitochondrial adaptor nucleoporin p62 in a human cell line [116]. Hypoxia regulates mitophagy through the HIF-1α–BNIP3 pathway in nucleus pulposus cells [122]. Mitochondria in nucleus pulposus cells undergo HIF-1α-dependent fragmentation by modulating dynamin-related protein 1 (DRP1) and mitochondrial dynamin-like GTPase [122]. HIF-1α positively regulates mitochondrial fission through DRP1 [123]. Similar to HIF-1α, mitophagy via BNIP3 signaling involves DRP1-mediated mitochondrial fission and recruitment of Parkin in cardiac myocytes [124]. Translocation of BNIP3 to the mitochondria causes mitochondrial depolarization by inducing the mitochondrial permeability transition pore (MPTP) [125]. This leads to cytosolic accumulation of mitochondrial molecules. The cytosolic accumulation of mtDNA or mtRNA triggers the activation of TANK-binding kinase 1 (TBK1) by phosphorylation, leading to the stimulation of interferon (IFN)-regulatory factor 3 (IRF3)-mediated immune activation via IFN1β, IL-6, and TNF [70]. Hyperglycemia may aggravate the complexity of coronary atherosclerosis by activation of TBK1–HIF-1α-mediated IL-17/IL-10 signaling in macrophages [126]. The TBK1 inhibitor reverses hyperglycemia-induced HIF-1α expression [126]. Thus, mitochondrial damage in macrophages may enhance HIF-1α expression, thereby promoting inflammation (Figure 2).

### 4.3. Role of HIF-1α in VDAC1-Mediated Mitochondrial Functions

Three voltage-dependent anion channel (VDAC) family members (VDAC1, VDAC2, and VDAC3) have been identified in mammalian mitochondria [127,128]. Among these three, a relationship between VDAC1 and HIF-1α has been reported [129]. HIF-1α and nuclear respiratory factor 1 can act as transcriptional activators of the *VDAC1* promoter following serum starvation and hypoxia [129]. Hypoxia induces ROS generation from the respiratory complex in the inner mitochondrial membrane [127]. ROS production and the consequent mtDNA release through VDAC1 oligomerization can facilitate apoptosis and inflammation. VDAC1 oligomerization into dimers, trimers, tetramers, and higher-order oligomers induces apoptosis by increasing mitochondrial outer membrane permeability, allowing the release of mtDNA into the cytoplasmic matrix [130,131]. In HeLa cells, inhibition of VDAC1 oligomerization reduced selenite-mediated cell apoptosis and mitochondrial dysfunction (i.e., cytochrome c release from the mitochondria to the cytosol, ROS levels, and decreased mitochondrial membrane potential) [132]. VDAC1 forms a macromolecule-sized pore in the outer membranes of the mitochondria, and its oligomerization mediates the transport of proteins and mtDNA [133] (Figure 2). A link between VDAC1 oligomerization and inflammation in inflammatory diseases has also been reported [127]. 

Mitochondrial HIF-1α plays a somewhat beneficial role in cell survival. Oxidative stress induces mitochondrial translocation of endogenous HIF-1α in HeLa cells [115]. Mitochondria-localized HIF-1α reduces oxidative stress and increases cell survival [115]. HIF-1α associated with the outer mitochondrial membrane protects the integrity of mitochondrial membrane potential and prevents apoptosis by directly regulating VDAC1 and hexokinase 2, leading to the production of a C-terminally truncated active form of VDAC1 [114]. 

HIF-1α modulates cell metabolism by hypoxia, regulating glucose transporter-1 and hexokinase 2 expression in various cell types [134,135,136]. Hexokinase 2 catalyzes the first stage of glycolysis and suppresses apoptosis by binding to VDAC on the mitochondrial membrane [137]. HIF-1α may be associated with the regulation of mitochondrial functions via direct interactions with hexokinase 2 [137]. A recent report showed that hexokinase 2 dissociation from VDAC triggers the activation of inositol triphosphate receptors, leading to the release of Ca^2+^ from the endoplasmic reticulum, which is taken up by mitochondria [138]. This influx of Ca^2+^ into the mitochondria leads to the oligomerization of VDAC, which facilitates NLRP3 inflammasome assembly and activation [138]. The relationship between Aβ-mediated toxicity and VDAC1 has been reported previously. While the VDAC1-N-terminal peptide shows protective effects against Aβ-mediated human neuroblastoma cell apoptosis, VDAC1 facilitates Aβ-mediated cell toxicity, demonstrating mitochondrial dysfunction and apoptosis induction [139]. VDAC1 inhibition enhances mitochondrial function and synaptic activity [140]. Hence, while mitochondrial HIF-1α may protect against oxidative stress, transcriptional activity of HIF-1α may enhance VDAC1 expression leading to VDAC1 oligomerization and consequent inflammasome formation.

## 5. Role of HIF-1α in Cellular Activation

HIF-1α is closely related to glial activation and consequent release of proinflammatory factors (Figure 3). Inflammatory factors induce BBB leakage by changing the structures of tight junction proteins [141,142]. Endothelial damage, pericyte apoptosis, reactive glial activation (gliosis), and inflammatory cytokines exacerbate CNS neurodegeneration by uncoupling normal cell–cell communication [9]. The infiltration of immune cells through leaky vessels further stimulates various brain cells located in the neurovascular unit.

### 5.1. Astrocyte Activation

Inactivation of astrocytic *VEGFA* expression reduces BBB leakage in inflammatory CNS diseases [143]. In the acute phase of stroke, excessive VEGF acts as a potent vascular permeability factor [17,141,144]. Ischemia/reperfusion-injury-mediated release of proinflammatory cytokines and other soluble mediators triggers paracellular permeability and tight junction disruption [145,146,147]. Tight junctions are disrupted during neuroinflammatory diseases, which results in the infiltration of monocytes into the brain parenchyma, where they become activated macrophages [147,148].

Cortical astrocytes located in the penumbra of an ischemic stroke rat model show enhanced levels of high mobility group box 1 (HMGB1) and its receptor TLR4 [149]. Administration of recombinant HMGB1 to the normal rat cortex triggers the expression of TLR4 and its downstream mediator, iNOS, in astrocytes [149]. HIF-1α-mediated human iNOS expression is seen in primary human astrocytes under cytokine-stimulated conditions [101]. HMGB1 is upregulated in human astrocytoma tissues. Moreover, hypoxia-induced HIF-1α is an upstream regulator of HMGB1 in human glioma stem cell lines [150]. iNOS-derived NO triggers the post-translational S-nitrosylation of HMGB1, leading to HMGB1 secretion and proinflammatory responses [151]. Secreted HMGB1 acts in a damage-associated molecular pattern (DAMP), activating the NLRP3 inflammasome [152,153]. Multiple inflammasome-related complications affect immune system homeostasis in patients with severe TBI [154].

### 5.2. Oligodendrocyte Activation

In the brain, oligodendrocytes produce myelin, which is a lipid-rich membrane. Oligodendrocytes in the white matter have a high metabolic demand that requires mitochondrial ATP production during remyelination processes [155]. White matter degeneration has been correlated with decreased cognitive function during normal brain aging [156]. Hypoxic oligodendrocyte precursor cell (OPC)-derived VEGF is associated with BBB impairment [157]. HIF-1α activates a unique set of genes in OPCs through interaction with the OPC-specific transcription factor OLIG2, which results in impaired oligodendrocyte formation [158]. The receptors for HMGB1 in OPCs include TLR2, TLR4, TLR9, and the receptor for advanced glycation end-product (RAGE) [159]. Treatment of OPCs with HMGB1 blocks OPC maturation into oligodendrocytes and triggers nuclear translocation of NF-κB through a TLR2-dependent pathway [159]. RAGE expression is influenced by hypoxia via nuclear translocation of NF-κB and HIF-1α [160]. 

### 5.3. Microglia/Macrophage Activation

The levels of mitochondrial DAMPs (i.e., mtDNA) in patients are often associated with the severity and prognosis of human diseases. Mitochondrial DAMPs are released into the extracellular space, causing immune responses [161]. Immune cells such as macrophages and microglia are activated under hypoxic conditions, leading to increased mobilization [162]. In human AD brains, endothelial cells upregulate genes involved in cytokine secretion and immune responses [29]. AD microglia downregulate homeostatic genes [29]. Inhibition of autophagy in microglia and macrophages exacerbates the innate immune responses and worsens brain injury outcomes [163]. Autophagic flux can be disrupted in brain cells following TBI in mice. Macrophage-/microglia-specific knockout of the essential autophagy gene beclin1 leads to an overall increase in neuroinflammation after TBI [163]. Increasing autophagy following rapamycin treatment decreases inflammation and improves the outcomes in wild-type mice after TBI [163]. 

In neuroinflammatory disease, increased HMGB1 expression can be detected in astrocytes, microglia, and infiltrating macrophages, which triggers the HMGB1–TLR4–NF-κB signaling pathway [164]. STAT3 activation in microglia may affect pericyte cell death [165]. STAT3 ablation in microglia induces pericyte survival in diabetic retinas via TNF-α–Akt–p70S6 kinase signaling [165]. Activated STAT3 increases HIF-1α protein stability and accelerates de novo synthesis of HIF-1α [118]. 

Depending on the disease stage and chronicity, microglia are stimulated differently, leading to particular activation states (M1 and M2), which correspond to altered microglial morphology, gene expression, and function [166]. Ramified microglial morphology (M2 phenotype) is associated with normal surveillance activity, while a more rounded phagocytic appearance (M1 phenotype) is observed in the damaged brain [166]. Macrophage-specific HIF-1α-deficient mice show suppressed wire-induced neointimal thickening and decreased infiltration of inflammatory cells as compared to wild-type mice [167]. This result implies that decreasing HIF-1α activity in macrophages may prevent the progression of vascular remodeling [167]. Additionally, HIF-1α-deficient macrophages are positively correlated with the phenotypic profile of M2 macrophages and negatively correlated with that of M1 macrophages [167]. 

### 5.4. Vascular Cells

When mice were exposed to chronic mild hypoxia (8% O_2_), leaky blood vessels were noted [168]. Prolonged hypoxia has deleterious effects on AD pathogenesis [1,14,169,170]. Microvessels obtained from the brains of patients with AD express higher levels of HIF-1α protein than do those in controls [171]. Plasma levels of HMGB1 increase within 30 min of severe trauma in humans, which correlates with tissue hypoperfusion [172].

In spinal cord injury models, primary mouse brain microvascular endothelial cells engulf myelin debris through immunoglobulin G opsonization [173]. Myelin debris in endothelial cells can then be delivered to the lysosomal degradation system via the autophagy pathway [173]. Autophagic degradation of myelin debris is required for endothelial cell proliferation, via VEGF [173]. The uptake of myelin debris by endothelial cells stimulates macrophage recruitment by upregulating monocyte chemoattractant protein-1 (MCP-1), inflammatory responses, and glial activation [173]. 

MCP-1 upregulates HIF-1α gene expression in human endothelial cells, resulting in VEGF induction [174]. In AD specimens, cerebral microvessels showed higher levels of MCP-1, IL-1β, IL-6, VEGF, and matrix metalloproteinase-9 (MMP-9) than those found in age-matched control brains [175,176,177]. MMPs are a family of zinc- and calcium-dependent endopeptidases degrading the extracellular matrix [175,178]. Among the MMPs, MMP-9 deficiency reduced BBB leakage following TBI in mice [179]. Animal models of TBI or stroke demonstrate higher levels of MMP-9 in the ipsilateral brain regions [9,179]. MMP-9 enables pericytes to detach from the basal lamina, migrate to the newly formed microvasculature, and balance the degradation and maturation of the vasculature after ischemic stroke [180,181]. HIF-1α induces vascular permeability during severe hypoxic conditions by increasing MMP-9 and VEGF expression [182].

## 6. Conclusions and Future Directions 

This review has revealed the molecular mechanisms of a key molecule, HIF-1α, during severe hypoxic conditions, such as those in brain diseases. Severe and chronic hypoxia exacerbates inflammation, mitochondrial malfunction, excessive oxidative stress, and cell death, partly due to the disproportionate accumulation of HIF-1α. 

Currently, most HIF-1α inhibitors have been tested in preclinical models of solid tumors [183]. There are currently no clinical trials using HIF-1α inhibitors in stroke, TBI, or AD (https://clinicaltrials.gov/ accessed on 8 April 2024). Instead, clinical trials using “hyperbaric oxygen treatment” are being carried out in mild TBI (NCT02089594, NCT01220713, NCT00594503), stroke (NCT04376359, NCT06148285, NCT04149379, NCT03431402), and mild cognitive impairment (NCT02085330). High O_2_ therapy can elevate cerebral blood flow and improve cognitive behavioral performances by diminishing hypoxia.

Developing techniques to diminish HIF-1α during severe hypoxia is valuable, creating a new direction for brain disease treatment. Proper inactivation of HIF-1α may contribute to the reduction in inflammasomes and cell damage and to enhanced mitochondrial function through the transcriptional regulations and post-modification of target molecules in neurodegenerative diseases, such as stroke, TBI, and AD. 

## Figures and Tables

**Figure 1 ijms-25-04465-f001:**
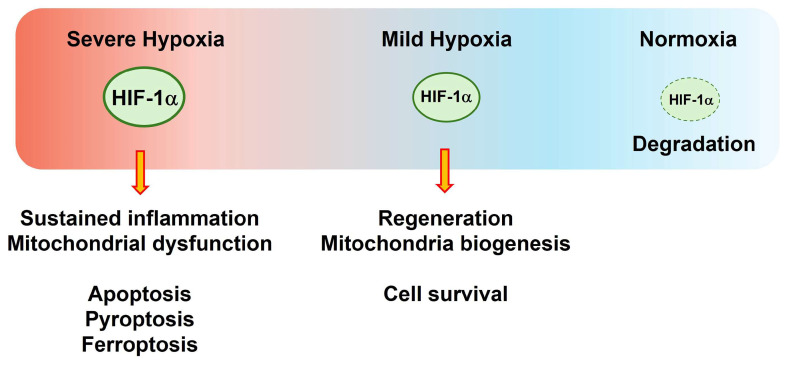
The roles of hypoxia-inducible factor (HIF)-1α in different oxygen tension conditions in the central nervous system (CNS). In severe hypoxic regions of the brain, HIF-1α can induce sustained inflammation and mitochondrial dysfunction, consequently leading to cell death in the form of apoptosis, pyroptosis, and ferroptosis. HIF-1α stabilization during mild hypoxia may enhance cell regeneration (i.e., angiogenesis and neurogenesis), mitochondrial biogenesis, and cell survival in the brain. During normoxia, HIF-1α can be degraded in the CNS.

**Figure 2 ijms-25-04465-f002:**
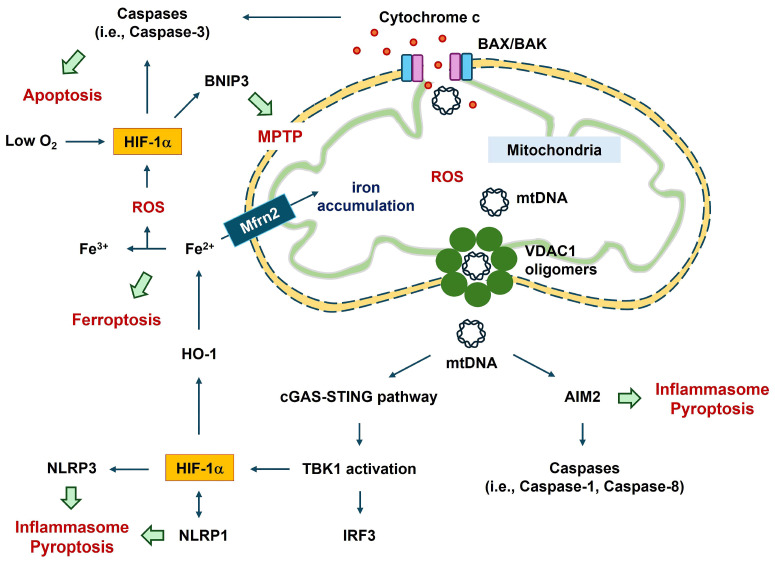
Involvement of HIF-1α in mitochondrial-mediated cell damage during severe hypoxia in the CNS. Mitochondria act as multifaceted regulators of cell death, and HIF-1α mediates MPTP, apoptosis, ferroptosis, inflammasome formation, and pyroptosis. Abbreviations: AIM2, absent in melanoma 2; BAX, BCL-2-associated X, apoptosis regulator; BAK, BCL-2 antagonist/killer 1; BNIP3, BCL-2/adenovirus E1B 19 kDa-interacting protein 3; cGAS, cytosolic cyclic GMP–AMP synthase; CNS, central nervous system; HIF-1α, hypoxia-inducible factor-1α; HO-1, heme oxygenase-1; IRF3, interferon regulatory factor; Mfrn2, mitoferrin 2; mtDNA, mitochondrial DNA; MPTP, mitochondrial permeability transition pore; NLRP1, NLR family pyrin domain containing 1; STING, stimulator of interferon genes; TBK1, TANK-binding kinase 1; ROS, reactive oxygen species; VDAC1; voltage-dependent anion channel 1.

**Figure 3 ijms-25-04465-f003:**
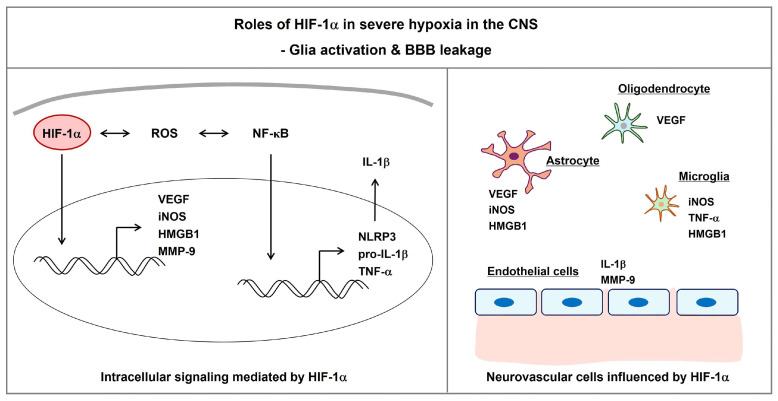
HIF-1α-mediated glial activation and BBB leakage during severe hypoxia in the CNS. In severe hypoxia, HIF-1α can regulate the release of cytokines and production of inflammatory mediators in endothelial cells and activated glial cells (i.e., astrocytes, oligodendrocytes, microglia), leading to enhanced BBB leakage. Abbreviations: BBB, blood–brain barrier; CNS, central nervous system; HMGB1, high-mobility group box 1; IL-1β, interleukin-1β; iNOS, inducible nitric oxide synthase; MMP-9, matrix metalloproteinase; NF-κB, nuclear factor-κB; NLRP3, NLR family pyrin domain-containing 3; ROS, reactive oxygen species; TNF-α, tumor necrosis factor-α; VEGF, vascular endothelial growth factor.

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
