# Peer review of "Detrimental Roles of Hypoxia-Inducible Factor-1α in Severe Hypoxic Brain Diseases"

_ijms, 2024, doi:10.3390/ijms25084465_

Round 1
Reviewer 1 Report
Comments and Suggestions for Authors
This is a review paper from Dr. Choi’s lab. The focus is to summarize the role of HIF in pathological conditions, especially in brain injury induced by ischemia-reperfusion, TBI, and AD. Dr. Choi discussed the HIF1 role in apoptosis, inflammation, mitochondrial dysfunction, and autophagy. He concluded that an increase in HIF1 content mainly exacerbates inflammation, mitochondrial malfunction, excessive oxidative stress, and cell death. Although Dr. Choi did point out the HIF1 may play protective role in certain conditions.
While it is a well-written manuscript, lack of details is a major weakness in the current version of manuscript.
2.1. Apoptosis
Author talked about the interaction between HIF-1a and apoptosis proteins including bcl-2, bax et al. However, the details were missing here. What are the potential binding sites between HIF1 and apoptosis proteins? Were post-translational modifications in apoptosis proteins required for HIF1 to bind with these proteins?
Line 106-108
“HO-1 resides within the endoplasmic reticulum, and its sustained expression in glia exacerbates AD development [40]. Prolonged expression of HO-1 in astrocytes leads to abnormal iron deposition in the mitochondria, resulting in neuronal deficiency and reduced cognitive ability [41-44].”
Again, more information is needed in this section. HO-1 exists in the ER of astrocytes. What type of mitochondria were affected by iron deposition? Mitochondria in the astrocytes or mitochondria in the neurons. If HO-1 expression only affected astrocytes, how this defect can cause neuronal deficiency and cognitive impairment?
Line 241
“When mitochondria sense hypoxia, it leads to mitochondrial ROS generation,--
Could author explain how mitochondria sense hypoxia and what is the potential mechanism to increase ROS generation?
Line 245-246
“Oxidative stress contributes to overproduction of ROS,---“
For my understanding, it is the overproduction of ROS leads to oxidative stress.
There were many studies and review papers about autophagy and mitophagy. Author did superficial discussion about autophagy and mitophagy. It is better to shorten the autophagy discussion in the revised manuscript.
Line 390
“VEGF-expressing astrocytes promote BBB leakage in the ischemic brain [135, 136].”
What did this sentence mean here?
Author mentioned the mtDNA or RNA in several sections. Could author summarize the role of mtDNA/RNA in one specific section to prevent redundancy?
Author Response
Reviewer #1
Comment #1
2.1. Apoptosis
Author talked about the interaction between HIF-1a and apoptosis proteins including bcl-2, bax et al. However, the details were missing here. What are the potential binding sites between HIF1 and apoptosis proteins? Were post-translational modifications in apoptosis proteins required for HIF1 to bind with these proteins?
Response #1: Overexpression of HIF-1a upregulated active monomeric form of BCL-2-associated X (BAX) in cardiomyocyte cells (Malhotra et al., 2008). The authors used the active monomer of BAX antibody (monoclonal antibody 6A7, 0.5 ug/ml concentration, Alexis Biochemicals) (Malhotra et al., 2008), this antibody recognized conformational changed active BAX. This structural change of BAX allows to translocate from the cytosol to the mitochondrial, which is a critical step in the activation of BAX and subsequent induction of apoptosis.
Another apoptotic factor is BCL-2/adenovirus E1B 19 kDa-interacting protein 3 (BNIP3). HIF-1a can bind on promoter region of BNIP3; thus, BNIP3 is a downstream target gene of HIF-1a.
According to your comment, I changed text.
Line 85-87: BCL-2/adenovirus E1B 19 kDa-interacting protein 3 (BNIP3), a downstream target gene of HIF-1a, is involved in apoptosis (Mitroshina et al., 2021; Zhang et al., 2007).
Line 101-104: Overexpression of HIF-1a upregulates active monomers of BAX in hypoxic cardiomyocyte cells (Malhotra et al., 2008). HIF-1a modulates apoptosis-inducing proteins (i.e., BAX, BNIP3), leading to mitochondrial dysfunction and consequent cell death under severe hypoxic conditions (Guo et al., 2001; Liao et al., 2023).
Also, I removed BAK from Line 100 because I could not find the interaction between HIF-1a and BAK. I am sorry for my misunderstanding.
Comment #2
Line 106-108
“HO-1 resides within the endoplasmic reticulum, and its sustained expression in glia exacerbates AD development [40]. Prolonged expression of HO-1 in astrocytes leads to abnormal iron deposition in the mitochondria, resulting in neuronal deficiency and reduced cognitive ability [41-44].”
Again, more information is needed in this section. HO-1 exists in the ER of astrocytes. What type of mitochondria were affected by iron deposition? Mitochondria in the astrocytes or mitochondria in the neurons. If HO-1 expression only affected astrocytes, how this defect can cause neuronal deficiency and cognitive impairment?
Response #2: (Schipper, 2011; Song et al., 2017; Zukor et al., 2009) The authors generated transgenic (Tg) mice that selectively express the Flag-tagged human HMOX1 gene within the astrocytic compartment. In the GFAP.hHO-1 mice (but not in wild-type littermates; Wt), co-expression of immunoreactive Flag and HO-1 was noted in glia (but not neurons) at embryonic day 12 and postnatal weeks 1, 6 and 48. Intriguingly, the Tg mice exhibited increased numbers of Perls-stained (iron-containing) glial inclusions in the amygdala (3.7-fold; p < 0.05) and substantia nigra (4.2-fold; p < 0.01) relative to Wt values.
Therefore, we added explanation according to your comments.
Line 112-120: HO-1 resides within the endoplasmic reticulum, and its sustained expression in glial fibrillary acidic protein (GFAP)-expressing astrocytes exacerbates AD development (Schipper, 2011). Transgenic mice exhibiting prolonged expression of HO-1 in astrocytes (GFAP.HMOX1 transgenic) acquire abnormal iron deposition in the mitochondria of astrocytes located in the striatum as well as neuronal deficiency and reduced cognitive ability (Choi and Kim, 2022; Schipper, 1999; Schipper et al., 2019; Song et al., 2017). When GFAP.HMOX1 transgenic mice-derived astrocytes and neurons are co-cultured, active caspase-3 (an apoptotic factor) is increased in transgenic-derived neurons compared with wild-type preparations (Song et al., 2017). These studies suggest that overexpression of HO-1 in astrocytes may give rise to neuronal dysfunction.
Comment #3
Line 241
“When mitochondria sense hypoxia, it leads to mitochondrial ROS generation,--
Could author explain how mitochondria sense hypoxia and what is the potential mechanism to increase ROS generation?
Response #3: Hypoxia elicits ROS accumulation due to the insufficient amount of electron recipient, O2, causing imbalanced electron transfer in the electron transport chain (Kung-Chun Chiu et al., 2019).
Thus, we added it in the manuscript.
Line 252-257: During hypoxia, ROS is generated due to mitochondrial depolarization especially through complex I and III, activation of xanthine oxidase, and NADPH oxidases at different oxygen levels in the brain (Chen et al., 2018). Mitochondria consume O2 for ATP production. Hypoxia elicits mitochondrial ROS accumulation due to the insufficient amount of electron supplied by O2, causing imbalanced electron transfer in the electron transport chain (Kung-Chun Chiu et al., 2019).
Comment #4
Line 245-246
“Oxidative stress contributes to overproduction of ROS,---“
For my understanding, it is the overproduction of ROS leads to oxidative stress.
Response #4: We changed that sentence per your comment. Thank you for your valuable comment. Also, we added RNS formation because of another reviewer’s comment.
Line 261-265: ROS and reactive nitrogen species (RNS) contribute to oxidative stress production (Wang and Michaelis, 2010). One RNS, peroxynitrite (ONOO-), can be formed by nitric oxide (NO) and O2- (Garry et al., 2015). Hypoxia-induced HIF-1a can upregulate inducible nitric oxide synthase (iNOS) expression (Lee et al., 2019), leading to NO production. Uncoupling of endothelial NOS produces O2- during hypoxia (Sharma et al., 2023).
Comment #6
Line 390
“VEGF-expressing astrocytes promote BBB leakage in the ischemic brain [135, 136].”
What did this sentence mean here?
Response #6: We added explanation per your valuable comment.
Line 407-409: Inactivation of astrocytic VEGFA expression reduces BBB leakage in inflammatory CNS diseases (Argaw et al., 2012). In the acute phase of stroke, excessive VEGF acts as a potent vascular permeability factor (Moon et al., 2021; Pedram et al., 2002; Zhang et al., 2000).
Comment #5
There were many studies and review papers about autophagy and mitophagy. Author did superficial discussion about autophagy and mitophagy. It is better to shorten the autophagy discussion in the revised manuscript.
Comment #7
Author mentioned the mtDNA or RNA in several sections. Could author summarize the role of mtDNA/RNA in one specific section to prevent redundancy?
Response #5 & 7: We made new section (4.1. Mitochondrial DNA) and combined autophagy-related mtDNA discussion.
Line 273-293: 4.1. Mitochondrial DNA
The released mtDNA, cytosolic double-stranded DNA, can act as a ligand for various detrimental signal sensors, activating an innate immune response in a caspase-independent manner (Perez-Trevino et al., 2020). These signal sensors include the NLRP3 inflammasome, AIM2 inflammasome, and the cytosolic cyclic GMP–AMP synthase (cGAS)-stimulator of interferon genes (STING) pathway (Marchi et al., 2023; Perez-Trevino et al., 2020; Sharma et al., 2019). cGAS-STING signaling cascades facilitate the mtDNA-mediated secretion of inflammatory cytokines and activate immune cells (Marchi et al., 2023; Perez-Trevino et al., 2020) (Figure 2). mtDNA can be oxidized by ROS to generate fragments (Lee and Han, 2017; Perez-Trevino et al., 2020). The binding of cytosolic oxidized mtDNA to the NF-κB-mediated NLRP3 inflammasome suggests a link between apoptosis and the inflammasome (Shimada et al., 2012). Autophagy proteins (i.e., microtubule-associated protein 1 light chain 3B [LC3B] and beclin1) regulate the innate immune response by inhibiting NLRP3 inflammasome-mediated mtDNA release, leading to the preservation of mitochondrial integrity (Nakahira et al., 2011). Autophagy is an evolutionarily conserved, lysosome-dependent mechanism through which eukaryotic cells eliminate potentially cytotoxic or superfluous materials from the cytoplasm, thereby maintaining homeostasis (Marchi et al., 2023). Depletion of autophagic proteins promotes the accumulation of dysfunctional mtDNA in the cytosol in response to lipopolysaccharides and ATP in macrophages (Nakahira et al., 2011). CoCl2-induced hypoxia upregulates the expression of autophagy-related genes (ATGs), such as BNIP3, BECN1, LC3, ATG5, and ATG7 (Zhou et al., 2018). HIF-1a inhibitors reduce hypoxic preconditioning-mediated enhancement of BNIP3 and beclin1 protein levels (Lu et al., 2018).
References
Argaw, A.T., Asp, L., Zhang, J., Navrazhina, K., Pham, T., Mariani, J.N., Mahase, S., Dutta, D.J., Seto, J., Kramer, E.G., et al. (2012). Astrocyte-derived VEGF-A drives blood-brain barrier disruption in CNS inflammatory disease. J Clin Invest 122, 2454-2468.
Chen, R., Lai, U.H., Zhu, L., Singh, A., Ahmed, M., and Forsyth, N.R. (2018). Reactive Oxygen Species Formation in the Brain at Different Oxygen Levels: The Role of Hypoxia Inducible Factors. Front Cell Dev Biol 6, 132.
Choi, Y.K., and Kim, Y.M. (2022). Beneficial and Detrimental Roles of Heme Oxygenase-1 in the Neurovascular System. Int J Mol Sci 23.
Garry, P.S., Ezra, M., Rowland, M.J., Westbrook, J., and Pattinson, K.T. (2015). The role of the nitric oxide pathway in brain injury and its treatment--from bench to bedside. Exp Neurol 263, 235-243.
Guo, K., Searfoss, G., Krolikowski, D., Pagnoni, M., Franks, C., Clark, K., Yu, K.T., Jaye, M., and Ivashchenko, Y. (2001). Hypoxia induces the expression of the pro-apoptotic gene BNIP3. Cell Death Differ 8, 367-376.
Kung-Chun Chiu, D., Pui-Wah Tse, A., Law, C.T., Ming-Jing Xu, I., Lee, D., Chen, M., Kit-Ho Lai, R., Wai-Hin Yuen, V., Wing-Sum Cheu, J., Wai-Hung Ho, D., et al. (2019). Hypoxia regulates the mitochondrial activity of hepatocellular carcinoma cells through HIF/HEY1/PINK1 pathway. Cell Death Dis 10, 934.
Lee, M., Wang, C., Jin, S.W., Labrecque, M.P., Beischlag, T.V., Brockman, M.A., and Choy, J.C. (2019). Expression of human inducible nitric oxide synthase in response to cytokines is regulated by hypoxia-inducible factor-1. Free radical biology & medicine 130, 278-287.
Lee, S.R., and Han, J. (2017). Mitochondrial Nucleoid: Shield and Switch of the Mitochondrial Genome. Oxid Med Cell Longev 2017, 8060949.
Liao, W., Wang, M., Wu, Y., Du, J., Li, Y., Su, A., Zhong, L., Xie, Z., Gong, M., Liang, J., et al. (2023). The mechanisms of Huangqi Guizhi Wuwu decoction in treating ischaemic stroke based on network pharmacology and experiment verification. Pharm Biol 61, 1014-1029.
Lu, N., Li, X., Tan, R., An, J., Cai, Z., Hu, X., Wang, F., Wang, H., Lu, C., and Lu, H. (2018). HIF-1alpha/Beclin1-Mediated Autophagy Is Involved in Neuroprotection Induced by Hypoxic Preconditioning. J Mol Neurosci 66, 238-250.
Malhotra, R., Tyson, D.W., Rosevear, H.M., and Brosius, F.C., 3rd (2008). Hypoxia-inducible factor-1alpha is a critical mediator of hypoxia induced apoptosis in cardiac H9c2 and kidney epithelial HK-2 cells. BMC Cardiovasc Disord 8, 9.
Marchi, S., Guilbaud, E., Tait, S.W.G., Yamazaki, T., and Galluzzi, L. (2023). Mitochondrial control of inflammation. Nat Rev Immunol 23, 159-173.
Mitroshina, E.V., Savyuk, M.O., Ponimaskin, E., and Vedunova, M.V. (2021). Hypoxia-Inducible Factor (HIF) in Ischemic Stroke and Neurodegenerative Disease. Front Cell Dev Biol 9, 703084.
Moon, S., Chang, M.S., Koh, S.H., and Choi, Y.K. (2021). Repair Mechanisms of the Neurovascular Unit after Ischemic Stroke with a Focus on VEGF. Int J Mol Sci 22.
Nakahira, K., Haspel, J.A., Rathinam, V.A., Lee, S.J., Dolinay, T., Lam, H.C., Englert, J.A., Rabinovitch, M., Cernadas, M., Kim, H.P., et al. (2011). Autophagy proteins regulate innate immune responses by inhibiting the release of mitochondrial DNA mediated by the NALP3 inflammasome. Nat Immunol 12, 222-230.
Pedram, A., Razandi, M., and Levin, E.R. (2002). Deciphering vascular endothelial cell growth factor/vascular permeability factor signaling to vascular permeability. Inhibition by atrial natriuretic peptide. J Biol Chem 277, 44385-44398.
Perez-Trevino, P., Velasquez, M., and Garcia, N. (2020). Mechanisms of mitochondrial DNA escape and its relationship with different metabolic diseases. Biochim Biophys Acta Mol Basis Dis 1866, 165761.
Schipper, H.M. (1999). Glial HO-1 expression, iron deposition and oxidative stress in neurodegenerative diseases. Neurotox Res 1, 57-70.
Schipper, H.M. (2011). Heme oxygenase-1 in Alzheimer disease: a tribute to Moussa Youdim. J Neural Transm (Vienna) 118, 381-387.
Schipper, H.M., Song, W., Tavitian, A., and Cressatti, M. (2019). The sinister face of heme oxygenase-1 in brain aging and disease. Prog Neurobiol 172, 40-70.
Sharma, B.R., Karki, R., and Kanneganti, T.D. (2019). Role of AIM2 inflammasome in inflammatory diseases, cancer and infection. Eur J Immunol 49, 1998-2011.
Sharma, P., Sri Swetha Victoria, V., Praneeth Kumar, P., Karmakar, S., Swetha, M., and Reddy, A. (2023). Cross-talk between insulin resistance and nitrogen species in hypoxia leads to deterioration of tissue and homeostasis. Int Immunopharmacol 122, 110472.
Shimada, K., Crother, T.R., Karlin, J., Dagvadorj, J., Chiba, N., Chen, S., Ramanujan, V.K., Wolf, A.J., Vergnes, L., Ojcius, D.M., et al. (2012). Oxidized mitochondrial DNA activates the NLRP3 inflammasome during apoptosis. Immunity 36, 401-414.
Song, W., Cressatti, M., Zukor, H., Liberman, A., Galindez, C., and Schipper, H.M. (2017). Parkinsonian features in aging GFAP.HMOX1 transgenic mice overexpressing human HO-1 in the astroglial compartment. Neurobiol Aging 58, 163-179.
Wang, X., and Michaelis, E.K. (2010). Selective neuronal vulnerability to oxidative stress in the brain. Front Aging Neurosci 2, 12.
Zhang, L., Li, L., Liu, H., Prabhakaran, K., Zhang, X., Borowitz, J.L., and Isom, G.E. (2007). HIF-1alpha activation by a redox-sensitive pathway mediates cyanide-induced BNIP3 upregulation and mitochondrial-dependent cell death. Free radical biology & medicine 43, 117-127.
Zhang, Z.G., Zhang, L., Jiang, Q., Zhang, R., Davies, K., Powers, C., Bruggen, N., and Chopp, M. (2000). VEGF enhances angiogenesis and promotes blood-brain barrier leakage in the ischemic brain. J Clin Invest 106, 829-838.
Zhou, J., Li, C., Yao, W., Alsiddig, M.C., Huo, L., Liu, H., and Miao, Y.L. (2018). Hypoxia-inducible factor-1alpha-dependent autophagy plays a role in glycolysis switch in mouse granulosa cells. Biol Reprod 99, 308-318.
Zukor, H., Song, W., Liberman, A., Mui, J., Vali, H., Fillebeen, C., Pantopoulos, K., Wu, T.D., Guerquin-Kern, J.L., and Schipper, H.M. (2009). HO-1-mediated macroautophagy: a mechanism for unregulated iron deposition in aging and degenerating neural tissues. Journal of neurochemistry 109, 776-791.
Reviewer 2 Report
Comments and Suggestions for Authors
In the manuscript entitled "Sinister roles of hypoxia-inducible factor-1α in severe hypoxic brain diseases", the author discusses the current evidence regarding the contribution of severe hypoxia to CNS injuries, with an emphasis on HIF-1-mediated pathways, presenting the molecular mechanisms by which HIF-1α is involved in the pathogenesis of CNS injuries, such as stroke, traumatic brain injury, and Alzheimer’s disease.
This review presents a good overview of modern knowledge about the roles of HIF-1α in various CNS diseases, and as such, it is important for all scientists who deal with this field, both at the basic and at the applied level. The discussion included adequate and contemporary literature, used in such a way as to indicate the possibility of manipulating the synthesis of HIF-1α in the treatment of numerous modern diseases. The structure of the manuscript is transparent and clear, which makes the manuscript usable and easy to follow.
Finally, I would like to address only a few minor comments.
The Figure 1 should be of better resolution.
Is it okay to title the paper with sinister roles of... given that the paper also mentions the protective (beneficial) roles of HIF-1α (line 348)? Please discuss.
Figure 3 is somewhat unintuitive, since it does not show the molecular interactions that lead to BBB leakage. Is it possible to rework it?
Line 442: de novo (Italic)
Author Response
Reviewer #2
Finally, I would like to address only a few minor comments.
Comment #1
The Figure 1 should be of better resolution.
Response #1: We changed it. Thank you for your valuable comment.
Comment #2
Is it okay to title the paper with sinister roles of... given that the paper also mentions the protective (beneficial) roles of HIF-1α (line 348)? Please discuss.
Response #2: We changed the title as ‘detrimental roles of…’. In this review, I tried to collect detrimental roles of HIF-1a, and emphasize them.
I could not exclude the beneficial roles of HIF-1a even though this review focused on detrimental roles of HIF-1a. Thank you for your comment.
Comment #3
Figure 3 is somewhat unintuitive, since it does not show the molecular interactions that lead to BBB leakage. Is it possible to rework it?
Response #3: We changed it per your valuable comment.
Comment #4
Line 442: de novo (Italic)
Response #4: I changed it per your comment. Thank you for your suggestion.
Reviewer 3 Report
Comments and Suggestions for Authors
In the manuscript “Sinister roles of hypoxia-inducible factor-1 in severe hypoxic brain diseases”, the author discussed the current evidence regarding the contribution of severe hypoxia to CNS injuries, with an emphasis on HIF-1-mediated pathways. During severe hypoxia in the CNS, HIF-1 facilitates inflammasome formation, mitochondrial dysfunction, and cell death. The author discussed the molecular mechanisms of HIF-1. The author showed that proper inactivation of HIF-1 may contribute to the reduction of inflammasomes, cell damage, and mitochondrial function in neurodegenerative diseases. The topic of this manuscript is interesting and provide systemic knowledge and clues for future work.
1, The authors mentioned that under mild hypoxia HIF-1ɑ is beneficial to cell survival while under severe hypoxia HIF-1ɑ plays roles in sustained inflammation and mitochondrial dysfunction. I suggest the author would like to add one part to explain the mechanism between the two circumstances.
2. VDAC has several functions besides mtDNA migration. Does HIF-1ɑ modulate the other functions such as Ca++ flux?
3, Are there more clinical trial evidences of HIF-1ɑ effects on brain disease?
4, the title “Sinister roles of hypoxia-inducible factor-1 in severe hypoxic brain diseases”. “Sinister” is little bit strong or absolute. Is it possible to select one more appropriate term?
Comments on the Quality of English Languageok
Author Response
Reviewer #3
Comment #1, The authors mentioned that under mild hypoxia HIF-1ɑ is beneficial to cell survival while under severe hypoxia HIF-1ɑ plays roles in sustained inflammation and mitochondrial dysfunction. I suggest the author would like to add one part to explain the mechanism between the two circumstances.
Response #1: Thank you for your valuable comment. I tried to add some explanation between mild hypoxia vs. severe hypoxia.
Line 40-46: HIF-1a stabilization during mild hypoxia may enhance cell regeneration (i.e., angiogenesis and neurogenesis), mitochondrial biogenesis, and cell survival in the brain through HIF-1a target genes (Kim et al., 2020; Moon et al., 2021). In severe hypoxia, HIF-1a causes various gene expression changes and post-translational modifications related to cell damage, mitochondrial dysfunction, cellular lipid peroxidation, and inflammasome formation (Liang et al., 2023; Malhotra et al., 2008; Song et al., 2023; Wu et al., 2022; Yuan et al., 2021) (Figure 1). Here, this review focuses on the detrimental effects of HIF-1a on cell damage under severe hypoxic conditions.
HIF-1a has complex effects in the brain (Kim et al., 2020; Mitroshina et al., 2021; Moon et al., 2021); therefore, I could not contain sufficient mechanisms related to oxygen tension. If I have some opportunity to study about it, I would like to do it. I am sorry for that.
Comment #2. VDAC has several functions besides mtDNA migration. Does HIF-1ɑ modulate the other functions such as Ca++ flux?
Response #2: I could not find the role of HIF-1a in mitochondrial Ca2+ flux through VDAC located in the outer mitochondrial membrane. Instead, I found the relationship between the mitochondrial calcium uniporter (MCU) and HIF-1a in breast cancer.
The mitochondrial calcium uniporter regulates breast cancer progression via HIF-1α (Tosatto et al., 2016). Triple-negative breast cancer (TNBC) represents the most aggressive breast tumor subtype. Expression of the mitochondrial calcium uniporter (MCU) in the inner mitochondrial membrane (IMM), the selective channel responsible for mitochondrial Ca2+ uptake, correlates with tumor size and lymph node infiltration, suggesting that mitochondrial Ca2+ uptake might be instrumental for tumor growth and metastatic formation. Accordingly, MCU downregulation hampered cell motility and invasiveness and reduced tumor growth, lymph node infiltration, and lung metastasis in TNBC xenografts. In MCU-silenced cells, production of mitochondrial reactive oxygen species (mROS) is blunted and expression of the HIF-1a is reduced, suggesting a signaling role for mROS and HIF-1a, downstream of mitochondrial Ca2+. Finally, in breast cancer mRNA samples, a positive correlation of MCU expression with HIF-1a signaling route is present [Reviewed in (Delierneux et al., 2020)].
In this review, I did not include MCU discussion. I think further studies may be required to know the interplay between VDAC-mediated Ca2+ flux and HIF-1a.
Comment #3, Are there more clinical trial evidences of HIF-1ɑ effects on brain disease?
Response #3: I tried to add clinical trial evidence. I found that clinical trials using “hyperbaric oxygen treatment” are being carried out in CNS injuries. I added them in the manuscript.
Line 501-507: Currently, most HIF-1a inhibitors have been tested in preclinical models of solid tumors (Xu et al., 2022). There are currently no clinical trials using HIF-1a inhibitors in stroke, TBI, or AD (https://clinicaltrials.gov/). Instead, clinical trials using “hyperbaric oxygen treatment” are being carried out in mild TBI (NCT02089594, NCT01220713, NCT00594503), stroke (NCT04376359, NCT06148285, NCT04149379, NCT03431402), and mild cognitive impairment (NCT02085330). High O2 therapy can elevate cerebral blood flow and improve cognitive behavioral performances by diminishing hypoxia.
Comment #4, the title “Sinister roles of hypoxia-inducible factor-1 in severe hypoxic brain diseases”. “Sinister” is little bit strong or absolute. Is it possible to select one more appropriate term?
Response #4: “Sinister” was changed into “Detrimental”. Thank you for your valuable comment.
References
Delierneux, C., Kouba, S., Shanmughapriya, S., Potier-Cartereau, M., Trebak, M., and Hempel, N. (2020). Mitochondrial Calcium Regulation of Redox Signaling in Cancer. Cells 9.
Kim, S., Lee, M., and Choi, Y.K. (2020). The Role of a Neurovascular Signaling Pathway Involving Hypoxia-Inducible Factor and Notch in the Function of the Central Nervous System. Biomol Ther (Seoul) 28, 45-57.
Liang, Z., Zheng, Z., Guo, Q., Tian, M., Yang, J., Liu, X., Zhu, X., and Liu, S. (2023). The role of HIF-1alpha/HO-1 pathway in hippocampal neuronal ferroptosis in epilepsy. iScience 26, 108098.
Malhotra, R., Tyson, D.W., Rosevear, H.M., and Brosius, F.C., 3rd (2008). Hypoxia-inducible factor-1alpha is a critical mediator of hypoxia induced apoptosis in cardiac H9c2 and kidney epithelial HK-2 cells. BMC Cardiovasc Disord 8, 9.
Mitroshina, E.V., Savyuk, M.O., Ponimaskin, E., and Vedunova, M.V. (2021). Hypoxia-Inducible Factor (HIF) in Ischemic Stroke and Neurodegenerative Disease. Front Cell Dev Biol 9, 703084.
Moon, S., Chang, M.S., Koh, S.H., and Choi, Y.K. (2021). Repair Mechanisms of the Neurovascular Unit after Ischemic Stroke with a Focus on VEGF. Int J Mol Sci 22.
Song, W., Chen, Y., Qin, L., Xu, X., Sun, Y., Zhong, M., Lu, Y., Hu, K., Wei, L., and Chen, J. (2023). Oxidative stress drives vascular smooth muscle cell damage in acute Stanford type A aortic dissection through HIF-1alpha/HO-1 mediated ferroptosis. Heliyon 9, e22857.
Tosatto, A., Sommaggio, R., Kummerow, C., Bentham, R.B., Blacker, T.S., Berecz, T., Duchen, M.R., Rosato, A., Bogeski, I., Szabadkai, G., et al. (2016). The mitochondrial calcium uniporter regulates breast cancer progression via HIF-1alpha. EMBO Mol Med 8, 569-585.
Wu, Y., Wang, J., Zhao, T., Chen, J., Kang, L., Wei, Y., Han, L., Shen, L., Long, C., Wu, S., et al. (2022). Di-(2-ethylhexyl) phthalate exposure leads to ferroptosis via the HIF-1alpha/HO-1 signaling pathway in mouse testes. J Hazard Mater 426, 127807.
Xu, R., Wang, F., Yang, H., and Wang, Z. (2022). Action Sites and Clinical Application of HIF-1alpha Inhibitors. Molecules 27.
Yuan, D., Guan, S., Wang, Z., Ni, H., Ding, D., Xu, W., and Li, G. (2021). HIF-1alpha aggravated traumatic brain injury by NLRP3 inflammasome-mediated pyroptosis and activation of microglia. J Chem Neuroanat 116, 101994.
Reviewer 4 Report
Comments and Suggestions for Authors
This review entitled "Sinister roles of hypoxia-inducible factor-1α in severe hypoxic brain diseases", focuses on three CNS pathologies, has a highly molecular approach, based mostly on research work conducted on human subjects. It has a good writing of the introduction and the rest of the subtopics. Presents four basic subtopics: Role of HIF in cell damage (Apoptosis and Ferroptosis), Role of HIF in inflammasomes (focused on stroke, TBI and AD diseases), Role of HIF-1α in mitochondrial functions (with emphasis on the axis with BNIP3, and the participation of VDAC1), and Role of HIF-1α in cellular activation (based on Astrocyte, Oligodendrocyte and Microglia). Overall I think it is a good paper. My observations are as follows:
Abstract
The abstract and introduction should include a brief contextualization of the importance of understanding the relationship between hypoxia, HIFs, and CNS injury.
1. Introduction
Regarding paragraph on lines 56-59: why is it of great significance to understand the role of HIF-1α-mediated pathways in CNS injury?
Understanding the role of these pathways in severe hypoxic CNS injuries, such as ischemic stroke, TBI, and AD, is of great significance. HIF-1α-mediated pathways include neurovascular inflammation, oxidative stress, glial activation, vascular damage, mitochondrial dysfunction, and cell death (Figure 1) [1, 14, 26, 27].
2. Role of HIF in cell damage
Why don't you include the discussion of pyroptosis and necroptosis in this section?
Paragraph on lines 102-104: I find it a bit confusing. I suggest the following wording: “The Fenton reaction involves the production of oxygen free radicals (ROS) from the reaction between hydrogen peroxide (H2O2) and ferrous iron (Fe2+), which can trigger lipid peroxidation.”
Please provide a reference for the paragraph on lines 132-135.
4. Role of HIF-1 in mitochondrial functions
The main reactive oxygen species (ROS) that participate in this phenomenon of hypoxia and brain damage are not mentioned in the entire review. The role that reactive nitrogen species could play is not mentioned.
In Figure 2 only apoptosis and ferroptosis are indicated; why are necroptosis and pyroptosis not included?
5. Role of HIF-1α in cellular activation
Line 431: Autophagy in microglia and macrophages regulates the innate immune responses. Does the term "regulates" correspond to upregulates or downregulates?
6. Conclusion and future directions
The conclusion is very general. The wording of the conclusion should be improved.
Future directions are not mentioned by the authors.
Finally
Authors should include a section summarizing the clinical or therapeutic implications of understanding the molecular mechanisms by which HIF-1α is involved in the pathogenesis of CNS lesions, such as stroke, traumatic brain injury and Alzheimer's disease.
Author Response
Reviewer #4
Comment #1. Abstract
The abstract and introduction should include a brief contextualization of the importance of understanding the relationship between hypoxia, HIFs, and CNS injury.
Response #1: I added a brief contextualization of the importance of understanding the relationship between hypoxia, HIFs, and CNS injuries.
Line 19-20: Deciphering the molecular mechanisms of HIF-1a will contribute to the development of therapeutic strategies for severe hypoxic brain diseases.
Line 63-65: This review describes the important actions of HIF-1a in severe hypoxic CNS injuries and their potential pathogenetic mechanisms.
Comment #2. 1. Introduction
Regarding paragraph on lines 56-59: why is it of great significance to understand the role of HIF-1α-mediated pathways in CNS injury?
Understanding the role of these pathways in severe hypoxic CNS injuries, such as ischemic stroke, TBI, and AD, is of great significance. HIF-1α-mediated pathways include neurovascular inflammation, oxidative stress, glial activation, vascular damage, mitochondrial dysfunction, and cell death (Figure 1) [1, 14, 26, 27].
Response #2: I changed that sentence as it follows:
Line 61-61: Understanding the role of these pathways in severe hypoxic CNS injuries, such as ischemic stroke, TBI, and AD, will provide clues for therapeutic strategies.
Comment #3. 2. Role of HIF in cell damage
Why don't you include the discussion of pyroptosis and necroptosis in this section?
Response #3: The roles of HIF-1a in necroptosis were not found well. Therefore, I toned down about necroptosis in this review.
Pyroptosis is closely associated with inflammasomes. Thus, pyroptosis was not mentioned in section 2 because it was mentioned in inflammasome section.
Comment #4.
Paragraph on lines 102-104: I find it a bit confusing. I suggest the following wording: “The Fenton reaction involves the production of oxygen free radicals (ROS) from the reaction between hydrogen peroxide (H2O2) and ferrous iron (Fe2+), which can trigger lipid peroxidation.”
Response #4: Thank you very much for your comment. We changed it per your comment.
Line 108-110: The Fenton reaction involves the production of ROS from the reaction between H2O2 and ferrous iron (Fe2+), which can trigger lipid peroxidation.
Comment #5.
Please provide a reference for the paragraph on lines 132-135.
Response #5: That was my interpretation. I changed it.
Line 143-145: Taken together, HIF-1a may upregulate iron production and mitochondrial accumulation via HO-1. Additionally, HIF-a may be involved in ferritinophagy through the NCOA4-mediated pathway.
Comment #6. 4. Role of HIF-1a in mitochondrial functions
The main reactive oxygen species (ROS) that participate in this phenomenon of hypoxia and brain damage are not mentioned in the entire review. The role that reactive nitrogen species could play is not mentioned.
Response #6:
Line 261-265: ROS and reactive nitrogen species (RNS) contribute to oxidative stress production (Wang and Michaelis, 2010). One RNS, peroxynitrite (ONOO-), can be formed by nitric oxide (NO) and O2- (Garry et al., 2015). Hypoxia-induced HIF-1a can upregulate inducible nitric oxide synthase (iNOS) expression (Lee et al., 2019), leading to NO production. Uncoupling of endothelial NOS produces O2- during hypoxia (Sharma et al., 2023).
Comment #7. In Figure 2 only apoptosis and ferroptosis are indicated; why are necroptosis and pyroptosis not included?
Response #7: The roles of HIF-1a in necroptosis were not found well except bellowed:
Necroptosis mimics features of both apoptosis and necrosis. Necroptosis requires related proteins, such as receptor-interacting protein kinase-3 (RIPK3) and the effector mixed lineage kinase domain-like protein (MLKL) (Jiang et al., 2018). HIF-1a also regulates necroptosis-related proteins, such as RIPK3 and MLKL, in ischemic stroke (Jiang et al., 2018). Enhanced HIF-1a levels after ischemic stroke appear to be involved in RIPK3/MLKL activation, leading to activation of the NLRP3 inflammasome (Jiang et al., 2018).
Therefore, I toned down about necroptosis in this review by removing it in Figure 1.
Pyroptosis is closely associated with inflammasomes. I added pyroptosis in Figure 2.
Comment #8. 5. Role of HIF-1α in cellular activation
Line 431: Autophagy in microglia and macrophages regulates the innate immune responses. Does the term "regulates" correspond to upregulates or downregulates?
Response #8: We changed it per your valuable comment.
Line 448-449: Inhibition of autophagy in microglia and macrophages exacerbates the innate immune responses and worsens brain injury outcomes (Hegdekar et al., 2023).
Comment #9. 6. Conclusion and future directions
The conclusion is very general. The wording of the conclusion should be improved.
Future directions are not mentioned by the authors.
Comment #10. Finally
Authors should include a section summarizing the clinical or therapeutic implications of understanding the molecular mechanisms by which HIF-1α is involved in the pathogenesis of CNS lesions, such as stroke, traumatic brain injury and Alzheimer's disease.
Response #9 & #10: I added future directions and clinical implications regarding to HIF-1a. Thank you very much for your valuable comments.
Line 496-512:
- Conclusion and future directions
This review revealed the molecular mechanisms of a key molecule, HIF-1a, during severe hypoxic conditions, such as those in brain diseases. Severe and chronic hypoxia exacerbates inflammation, mitochondrial malfunction, excessive oxidative stress, and cell death, partly due to the disproportionate accumulation of HIF-1a.
Currently, most HIF-1a inhibitors have been tested in preclinical models of solid tumors (Xu et al., 2022). There are currently no clinical trials using HIF-1a inhibitors in stroke, TBI, or AD (https://clinicaltrials.gov/). Instead, clinical trials using “hyperbaric oxygen treatment” are being carried out in mild TBI (NCT02089594, NCT01220713, NCT00594503), stroke (NCT04376359, NCT06148285, NCT04149379, NCT03431402), and mild cognitive impairment (NCT02085330). High O2 therapy can elevate cerebral blood flow and improve cognitive behavioral performances by diminishing hypoxia.
Developing techniques to diminish HIF-1a during severe hypoxia is valuable, creating a new direction for brain disease treatment. Proper inactivation of HIF-1a may contribute to the reduction of inflammasomes, cell damage, and enhanced mitochondrial function through the transcriptional regulations and post-modification of target molecules in neurodegenerative diseases, such as stroke, TBI, and AD.
References
Garry, P.S., Ezra, M., Rowland, M.J., Westbrook, J., and Pattinson, K.T. (2015). The role of the nitric oxide pathway in brain injury and its treatment--from bench to bedside. Exp Neurol 263, 235-243.
Hegdekar, N., Sarkar, C., Bustos, S., Ritzel, R.M., Hanscom, M., Ravishankar, P., Philkana, D., Wu, J., Loane, D.J., and Lipinski, M.M. (2023). Inhibition of autophagy in microglia and macrophages exacerbates innate immune responses and worsens brain injury outcomes. Autophagy 19, 2026-2044.
Jiang, Q., Stone, C.R., Geng, X., and Ding, Y. (2018). Hypoxia-inducible factor-1 alpha and RIP3 triggers NLRP3 inflammasome in ischemic stroke. Brain Circ 4, 191-192.
Lee, M., Wang, C., Jin, S.W., Labrecque, M.P., Beischlag, T.V., Brockman, M.A., and Choy, J.C. (2019). Expression of human inducible nitric oxide synthase in response to cytokines is regulated by hypoxia-inducible factor-1. Free radical biology & medicine 130, 278-287.
Sharma, P., Sri Swetha Victoria, V., Praneeth Kumar, P., Karmakar, S., Swetha, M., and Reddy, A. (2023). Cross-talk between insulin resistance and nitrogen species in hypoxia leads to deterioration of tissue and homeostasis. Int Immunopharmacol 122, 110472.
Wang, X., and Michaelis, E.K. (2010). Selective neuronal vulnerability to oxidative stress in the brain. Front Aging Neurosci 2, 12.
Xu, R., Wang, F., Yang, H., and Wang, Z. (2022). Action Sites and Clinical Application of HIF-1alpha Inhibitors. Molecules 27.